# Pyrimidine Schiff Bases: Synthesis, Structural Characterization and Recent Studies on Biological Activities

**DOI:** 10.3390/ijms25042076

**Published:** 2024-02-08

**Authors:** Iwona Bryndal, Marcin Stolarczyk, Aleksandra Mikołajczyk, Magdalena Krupińska, Anna Pyra, Marcin Mączyński, Agnieszka Matera-Witkiewicz

**Affiliations:** 1Department of Organic Chemistry and Drug Technology, Faculty of Pharmacy, Wroclaw Medical University, 211A Borowska, 50-556 Wrocław, Poland; marcin.stolarczyk@umw.edu.pl (M.S.); marcin.maczynski@umw.edu.pl (M.M.); 2Screening Biological Activity Assays and Collection of Biological Material Laboratory, Wroclaw Medical University, 211A Borowska, 50-556 Wrocław, Poland; aleksandra.mikolajczyk@umw.edu.pl (A.M.); magdalena.krupinska@umw.edu.pl (M.K.); agnieszka.matera-witkiewicz@umw.edu.pl (A.M.-W.); 3Faculty of Chemistry, University of Wroclaw, 14 Joliot-Curie, 50-383 Wrocław, Poland; anna.pyra@uwr.edu.pl

**Keywords:** anticancer activity, cytotoxic activity, pyrimidine Schiff bases, X-ray structures, ADME prediction analysis

## Abstract

Recently, 5-[(4-ethoxyphenyl)imino]methyl-N-(4-fluorophenyl)-6-methyl-2-phenylpyrimidin-4-amine has been synthesized, characterized, and evaluated for its antibacterial activity against *Enterococcus faecalis* in combination with antineoplastic activity against gastric adenocarcinoma. In this study, new 5-iminomethylpyrimidine compounds were synthesized which differ in the substituent(s) of the aromatic ring attached to the imine group. The structures of newly obtained pyrimidine Schiff bases were established by spectroscopy techniques (ESI-MS, FTIR and ^1^H NMR). To extend the current knowledge about the features responsible for the biological activity of the new 5-iminomethylpyrimidine derivatives, low-temperature single-crystal X-ray analyses were carried out. For all studied crystals, intramolecular N–H∙∙∙N hydrogen bonds and intermolecular C–H∙∙∙F interactions were observed and seemed to play an essential role in the formation of the structures. Simultaneously, their biological properties based on their cytotoxic features were compared with the activities of the Schiff base (III) published previously. Moreover, computational investigations, such as ADME prediction analysis and molecular docking, were also performed on the most active new Schiff base (compound **4b**). These results were compared with the highest active compound **III**.

## 1. Introduction

Pyrimidine is widespread in nature as a component of nucleic acids (cytosine, thymine, and uracil) and many other natural and synthetic compounds, including drugs, and it has become known over time as a potent pharmacophore [1,2]. On the other hand, pyrimidine Schiff bases belong to the class of compounds that possess an imine or azomethine functional group and were described for the first time in 1864 [3]. It still arouses interest of scientists because of its wide range of pharmacological activities, including, among others, antimicrobial [4,5,6,7], anticancer [8,9], anti-inflammatory [10,11] or analgesic [12] properties.

Generally, imines are formed in the condensation of primary amines with carbonyl compounds followed by an elimination of water. Many methods of synthesis have been described, the most common being water removal [13,14,15,16], adding catalytic amounts of acids [17,18,19], using Lewis acids as catalysts [20,21,22,23], and irradiation techniques [24,25,26,27]. The synthesis of Schiff bases containing the pyrimidine scaffold has been also studied. Pyrimidine exhibits wide occurrence in nature as a constituent of nucleic acids, thymine, and many other natural and synthetic compounds including drugs [28,29,30], and over time it has become known as an effective pharmacophore. In the most common cases, the amino group attached to the pyrimidine ring is exploited to form the imine group in reaction with aldehydes [31,32,33,34,35], and in some cases pyrimidine-5-carbaldehyde is also used [36,37,38,39].

In our previous studies, we observed that 5-hydroxymethylpyrimidine with a tetrasulphide bridge at the 4-position has considerable antibacterial and antifungal properties [40]. In particular, the hydroxylation of 4-[(4-chlorobenzyl)sulphanyl]-5,6-dimethyl-2-phenylpyrimidine to its 5-hydroxymethyl derivative (denoted here as **I**, see Figure 1) significantly enhances cytotoxicity against cancer cell lines (HeLa, K562, and CFPAC) while simultaneously manifesting low toxicity against a normal (HUVEC) cell line [41]. Furthermore, pyrimidines possessing a 4-benzylsulphanyl group exhibit stronger toxicity than their 4-amino analogues [42].

During our further research on 6-methyl-2-phenylpyrimidin-4-amine, we mainly focused on its 5-aminomethyl derivatives (Figure 1), especially 5-[(4-ethoxyanilino)methyl]-N-(4-fluorophenyl)-6-methyl-2-phenylpyrimidin-4-amine (denoted as **II**) and its Schiff base, 5-[(4-ethoxyphenyl)imino]methyl-N-(4-fluorophenyl)-6-methyl-2-phenylpyrimidin-4-amine (denoted as **III**), obtained as the main condensation product of *p*-phenethidine with a pyrimidine-5-carbaldehyde derivative [43]. Both of these compounds differ in their conformation and mode of interactions—the molecules are linked into chains by intermolecular hydrogen bonds of N–H∙∙∙N or C–H∙∙∙O, respectively, in the amine or imine analogue. They also differ in their biological properties. Biological studies have shown that the presence of a -C=N- bond in position 5 of the 6-methyl-2-phenylpyrimidine-4-amine core increases its activity. Compound **III** can be considered a selective antibacterial agent against *Enterococcus faecalis* (MIC—16 µg/mL, MBC—32 µg/mL), combined with its anticancer effect against gastric adenocarcinoma (IC_50_ = 53.02 µM for AGS) [43].

Based on the above information and considering the need to discover and develop biological active agents, we synthesized new 6-methyl-2-phenylpyrimidin-4-amine derivatives by a structural modification of the 5-[(4-ethoxyanilino)methyl]-N-(4-fluorophenyl)-6-methyl-2-phenylpyrimidin-4-amine molecule (**III**) [43], its structure constituting a valuable core for further optimization. The structural elements of this molecule were exchanged in the following directions and only in the 5-position of the pyrimidine ring: a change of an amino group into an imine moiety in the functional group, a change of the substituent and/or the introduction of an additional substituent with different electro-/nucleophilic properties to the aryl ring connected to the imine group (Figure 1). The structures of the obtained compounds, four novel pyrimidine Schiff bases, 5-iminomethyl-6-methyl-2-phenylpyrimidine derivatives (denoted as **4a**–**d**), bearing potentially biologically active functionalities, were established by spectroscopy techniques and studied for their cytotoxic activity in vitro towards normal (RPTEC) and cancer (AGS, HeLa, HepG2, A172, Ca-co-2) cell lines. In order to expand current knowledge about the features responsible for the biological activity of the new 5-iminomethylpyrimidine derivatives, X-ray single-crystal and ADME prediction analyses were carried out, and their results are presented here.

## 2. Results and Discussion

### 2.1. Chemistry

Four pyrimidine Schiff bases of 5-iminomethyl-6-methyl-2-phenylpyrimidine derivatives, **4a**–**4d**, which differ in the substituent(s) of the aromatic ring attached to the imine group, were obtained as shown in Figure 2. Details for the synthesis leading to the obtainment of the starting aldehyde (**3**), namely 4-(4-fluoroanilino)-6-methyl-2-phenylpyrimidine-5-carbaldehyde, have been given by the authors previously [43]. The final Schiff bases, **4a**–**d**, were obtained by coupling the aldehyde (**3**) with aromatic amines in THF in the presence of a catalytic amount of indium(III) trifluoromethanesulphonate (In(OTf)_3_) as a catalyst and were purified by column chromatography; they were air-stable in their solid state. All newly obtained Schiff bases were characterized by MS, 1D NMR (^1^H, ^13^C and ^19^F), and IR spectroscopies (see Appendix A of this paper).

### 2.2. X-ray Structural Studies

X-ray diffraction analysis was used to determine the structure of the newly obtained compounds **4a**–**d**. As a result of the recrystallization of compounds **4a**–**d** from a mixture of ethanol and 2-propanol (1:1 *v*/*v*), yellow needle- or block-shaped crystals, suitable for single-crystal X-ray diffraction, were obtained (Table 1).

It has been established that the compounds **4a** and **4b** crystallize in the orthorhombic system; the structures are chiral and have the space group *Pna*2_1_ and *P*2_1_2_1_2_1_, respectively. The compounds **4c** and **4d** belong to the monoclinic crystal system and adopt different symmetries, i.e., *C*2/*c* and *P*2_1_/*c*, respectively. Three of the studied compounds comprise one crystallographically independent molecule in the asymmetric unit (Figure 1). Only compound **4c** has two molecules in the asymmetric unit, additionally disordered in two positions with an occupancy factor of 0.5. In both cases, the disordered part includes atoms from the (4-fluorophenyl)amino group attached to the C4 atom of the pyrimidine ring (disordered atoms are denoted as 42C-43C, 45C-46C in A and as 42D-46D and F4D in B). Moreover, in molecule B, the carbon atom of the methoxy group is also disordered in two positions, denoted as C58B and C58D (Figure 2).

All molecules of studied compounds contain the same 6-methyl-2-phenylpyrimidine-4-amino core and an N-(4-fluorophenyl) group in the 4-position, but differ in the 5-position of the pyrimidine ring, namely the substituent(s) aromatic ring attached to the imine group. Even though such a small change in the structure of the molecule is introduced, it significantly affects the molecule’s conformation. Selected values of geometric parameters allowing us to observe these changes in conformation are listed in Table 2.

On the whole, the molecule adopts the *trans* (*E*) configuration around the imine functional group in all cases. The pyrimidine ring is planar, and the phenyl group at the 2-position is almost coplanar with the plane of this ring in **4b**–**4d,** or only slightly twisted regarding the plane of the ring in **4a**, as also observed in the recently described Schiff base structure (**III**) [43]. Compounds **4a** and **III** differ only in their aromatic ring substituent (compound **4a** with F and compound **III** with OC_2_H_5_ [43]), which is probably the cause of their similar conformation. It should be noted that, in the structure of **4b**, the aryl substituent in the 5-position is the most twisted relative to the pyrimidine ring plane, by almost 42°. In the other cases, the conformation is almost syn-planar, with the greatest interplanar angle distortions involving the aryl substituent at the 5-position in the 5 to 15° range. Simultaneously, in the case of **4b**, the aryl substituent in the 4-position is the most coplanar with the pyrimidine ring plane compared to other tested compounds (Table 2).

In the structures of **4a**–**d**, one type of intramolecular hydrogen bond contact is formed, i.e., N–H∙∙∙N, with the graph-set S(6) [44] (Table 3, Figure 1 and Figure 2), which is a characteristic feature for the previously reported 6-methyl-2-phenylpyrimidin-4-amines [42] and also for 5-[(4-ethoxyphenyl)imino]methyl-N-(4-fluorophenyl)-6-methyl-2-phenylpyrimidin-4-amine (**III**) and its amine analogue (**II**) [43]. Additionally, the distance values for N4∙∙∙N5 are similar to those presented previously [43], and they are in the range of 2.651–2.692 Å.

The molecular packing in the crystalline state for compound **4a** is mainly determined by the intermolecular C–H∙∙∙F, which engages the fluorine atom of the (4-fluorophenyl)amino group as a double acceptor, and the H atoms of the aryl substituent at the 5-position and the methyl group, respectively. In this way, a one-dimensional hydrogen-bonded chain is formed (Figure 3). It is noteworthy that the fluorine atom of the aryl substituent in the 5-position does not participate in the formation of C–H∙∙∙F interactions, which results from the fact that the calculated H∙∙∙F distances are longer than the sum of H and F van der Waals radii [45].

In the crystal structure of **4b**, two types of intermolecular hydrogen bonds are recognized. The molecules of **4b** are linked by intermolecular C–H∙∙∙F interactions, which lead to the formation of a chain. The fluorine atom of the (4-fluorophenyl)amino group forms a hydrogen bond (as an acceptor) with the methyl group at the 6-position of the pyrimidine ring (as a donor). Such chains interact further via C–H∙∙∙Cl hydrogen bonds to form a layered structure, with the chlorine atom of the 3-chlorophenyl group acts as a double acceptor in these interactions (Figure 4).

The crystal structure of **4c** displays a hydrogen-bonded arrangement which is dominated by C–H∙∙∙F hydrogen bonds (Figure 5). Interestingly, even though compounds **4c** and **III** [43] contain an alkoxy group at the aromatic ring (compound **4c** with OCH_3_ (*ortho*) and compound III with OC_2_H_5_ (*para*) [43]), the crystal structure of III contains intermolecular C–H∙∙∙O interactions, while the structure of 4c does not. The molecules denoted as B(D) of compound **4c** are connected by intermolecular C–H∙∙∙F interactions, which leads to the formation of a dimer in which (4-fluorophenyl)amine groups act as donors and acceptors. The molecules denoted as A are connected to such dimers through other intermolecular C–H∙∙∙F con-tacts mainly involving methyl groups (Table 3, Figure 5).

In the crystal structure of **4d**, two chains are formed, both due to hydrogen bonding of the C–H∙∙∙F type. Similar to other studied compounds, in each of these chains, the fluorine atom of a (4-fluorophenyl)amino group serves as an acceptor of the aryl H atom. On the other hand, the fluorine atom of a -CF_3_ group acts as an acceptor to the aryl H atoms at the 5-positon, resulting in a ribbon of molecules (Figure 6).

### 2.3. Biological Activity Analysis

#### 2.3.1. Neutral Red Uptake Assay

In order to evaluate the cytotoxic properties of the tested compounds (**4a**–**d**), a neutral red uptake (NR) assay was performed on the RPTEC cell line. Four concentrations were used: 10, 100, 250, and 500 µM of each compound. The obtained results are shown in Figure 7. The compound **4a** was rejected from further investigation on cancer cell lines due to its high cytotoxicity (53 and 67% at 100 and 250 µM, respectively). Furthermore, at a concentration of 500 µM, the compound **4a** crystallized, which influenced the obtained results by attenuating cytotoxic activity.

Based on the results obtained for the RPTEC cell line, the following concentrations of **4b**–**d** were chosen to be applied to neoplastic cell lines: 0.1, 1, 10, 25, 50, and 100 µM. A172, AGS, CaCo-2, HeLa, and HepG2 cancer cell lines were used to perform a neutral red uptake assay. The obtained results were compared with a negative control (cell lines incubated without tested compounds), and half-maximal inhibitory concentration (IC_50_) values were calculated. The results are presented in Table 4.

Only compound **4b** was cytotoxic enough to decrease the cell viability of the A172 and AGS cell lines to 50% in the tested concentration range, although it did not exert cytotoxic activity on CaCo-2, HeLa, or HepG2 cell lines. Compounds **4c** and **4d** were not cytotoxic in this concentration range. The calculated IC_50_ values were 63.385 µM and 32.210 µM for A172 and AGS, respectively, and they were used to investigate further details considering the mechanisms of cell death.

#### 2.3.2. Flow Cytometry Analysis

A detailed analysis of potential cytotoxic mechanisms has been performed only for the selected compound (**4b**) where the IC_50_ values under 100 µM were detected in the screening NR assay. Flow cytometry was performed on the A172 and AGS cell lines. Cells were incubated with compound **4b** at IC_50_ for 24, 48, and 72 h before analysis. Fluorescein diacetate (FDA) and propidium iodide (PI) were used as indicators for viable and nonviable (necrotic) cells. The results are shown in Figure 8. Most of the events were captured in the first and the fourth quadrant, which represent viable and necrotic cells, respectively. For the A172 line (Figure 8A), the viability of cells decreased to 59.32%/50.98%/43.88% after 24/48/72 h, respectively. Similar results were obtained for the AGS cell line (Figure 8B), where the viability was 60.95%/55.10%/44.99% after the same incubation periods, respectively. Some events occurred in the second quadrant, but it was difficult to unequivocally categorize them into viable or nonviable (apoptotic/necrotic) cell populations, and thus more sensitive dyes, able to distinguish between the types of cell death, were needed.

To confirm the cell death pathway, another approach to flow cytometry was used. Cells, after 6, 18, and 24 h incubation with compound **4b** at IC_50_, were stained with propidium iodide (PI) as in the previous experiment, but also with annexin V. One of the main characteristics of apoptotic cells is the phosphatidylserine (PS) residues’ translocation from the inner to the outer membrane of the cytoplasmic membrane. Annexin V is a specific PS-binding protein, and thus it is used to detect apoptotic cells. The results are presented in Figure 9. Very few events could be observed in the first and the second quadrant, which represent apoptotic cells. Most events occurred in the third and the fourth quadrant, where unstained and necrotic cells were located. The 24 h measurements after both flow cytometry approaches were consistent with each other. Thus, it can be concluded that compound **4b** activity leads to irreversible cell injury and necrosis in both tested cell lines.

#### 2.3.3. Cell Morphology

In order to obtain information about cells and nuclei morphology, fluorescence microscopy was used. Briefly, after the incubation of A172 and AGS with compound **4b** (at three time periods, namely 24, 48, and 72 h) cells were stained with Hoechst 33342 dye, which binds to DNA. Phase-contrast observations were also performed. The results are presented in Figure 10. The nuclei of healthy proliferating cells should be spherical, with DNA evenly distributed in it, as can be seen in A1,2 (A172) and B1,2 (AGS), which depict control cells incubated with 1% DMSO for 72 h. Also, the cell membrane should be intact with a visible halo around the cells. In necrotic cells, the nuclei edges tend to be less clearly defined, and Hoechst 33342 results in a decreased fluorescence signal. This effect can be seen in a1,2 (A172) and b1,2 (AGS). Overall, the microscopic observations confirmed the results obtained during flow cytometry, that the compound **4b** leads to necrosis in the observed cell lines.

#### 2.3.4. Genotoxicity Assay

Single-cell gel electrophoresis was performed to assess the potential genotoxic effect on AGS and A172 after incubation with compound **4b** at IC_50_. This method is based on the ability of damaged DNA to migrate out of the cell during electrophoresis. After DNA staining (Hoechst 33342), the characteristic “comet tail” could be observed under the fluorescence microscope. The damages in 100 randomly selected cells were visually classified into five groups according to the length of the “comet tail”. Group 0 consisted of cells without damaged DNA, while group 4 contained cells with maximally long tails. Based on this, a group damage index (DI) was calculated. The results are presented in Figure 11. The genotoxic effect was found in both examined cell lines. In the case of the A172 cell line, the DNA degradation level seemed to be independent of the incubation time with compound **4b** (DNA damage index (DDI) between 77 and 85 over three incubation periods), whereas for AGS cells, genotoxicity was induced in a time-dependent manner (DDI 63, 107, and 111 after 24, 48, and 72 h, respectively).

#### 2.3.5. Antimicrobial Activity Assay

In vitro antimicrobial and antifungal activity was screened using microdilution and modified Richard’s medium. The assay was performed for compounds **4a**, **4b**, **4c,** and **4d** with seven reference microbial strains from ATTC (*E. coli* 25922, *S. aureus* 43300, *K. pneumoniae* 700603, *A. baumannii* 19606, *P. aeruginosa* 27853, *E. faecalis* 29212, and *C. albicans* 10231). Neither MIC nor MBC/MFC activity was observed in the concentration range from 0.5 µg/mL to 256 µg/mL.

### 2.4. In Silico Studies

#### 2.4.1. ADME Prediction Analysis

Of the newly obtained pyrimidine Schiff bases, only compound **4b** was selected for in silico analysis due to its potential anticancer activity. Furthermore, compound **III** [43] was included in these studies for comparison purposes, as was *N*-1*H*-indazol-5-yl-2-(6-methylpyridin-2-yl)quinazolin-4-amine, which, according to the SwissParam database, was found to have similarity score of 0.530 relative to compound **4b** [46]. That found compound belongs to the class of organic compounds known as pyridinylpyrimidines. The results of the ADME prediction analysis for the selected compounds are summarized in Table 5.

#### 2.4.2. Molecular Docking Analysis

Molecular docking is an increasingly important tool in drug discovery and can be mainly used to represent the interaction between the amino acid residue of a protein and a small molecule of a ligand at the atomic level. In the present study, a model of the interaction between molecules and a receptor at the atomic level was designed in order to characterize the behaviour of molecules **III** and **4b** at the binding site of a target protein as well as to elucidate fundamental biochemical processes. The target prediction results are shown in Table 6. Docking was performed for many targets; however, the results shown here involve the one target for which the docking results were the best.

Vascular endothelial growth factor receptor 2 (VEGFR2) is highly expressed in several solid tumours and plays an important role in the apoptosis process. VEGFR2 inhibition has emerged as a promising approach for developing new therapies for many apoptosis-dependent cancers. Figure 12 and Figure 13 present the hydrogen bonds formed between the receptor and ligand (compound **III**) and the visualisation from MGLTools software 1.5.6 [67]. Figure 14 and Figure 15 present the actual ligand (compound **4b**) arrangement in its best conformation pose with the VEGFR2.

Docking studies revealed that compound **4b** showed better binding affinity to VEGFR2 (−8.2 kcal/mol) than Schiff base **III** [43] (−6.8 kcal/mol) in their best ranked conformation. In-depth analysis revealed that compound **4b** formed hydrogen bonds with Cys114, Phe113, Glu80, and Glu112 of VEGFR2, while recently reported compound **III** [43] formed hydrogen bonds with the same residues and additionally with Asp191 of VEGFR2. Due to the presented possible mode of action, compound **4b** may be a valuable drug candidate.

## 3. Materials and Methods

### 3.1. Chemistry

The reagents were purchased and used without purification. Lithium aluminium hydride, reagent grade 95%; pyridinium chlorochromate, 98%; thionyl chloride; silica gel, 200–400 mesh, 60 Å for column chromatography; CDCl_3_ for NMR spectroscopy; and amines were supplied by Sigma Aldrich, Darmstadt, Germany. Other reagents were provided by Chempur, Piekary Śląskie, Poland. TLC sheets Alugram SIL G/UV254 were obtained from Mecherey-Nagel, Germany.

NMR spectra were recorded on a Bruker ARX 300 MHz NMR Spectrometer. Chemical shifts (δ in ppm) were obtained from internal solvent-CDCl_3_ 7.26 ppm for ^1^H. ^13^ C and ^19^F NMR spectra were acquired on a Bruker Avance III 600 MHz spectrometer (Bruker Biospin GmbH, Ettlingen, Germany). Abbreviations used in NMR spectra: s—singlet, d—doublet, t—triplet, q—quartet, sx—sextet, m—multiplet. HR-MS spectra were recorded on a Bruker Daltonic model Compact, using the ESI technique. IR spectra were recorded on a Thermo Scientific Inc. (Waltham, MA, USA) model Nicolet iS50 FT-IR using the ATR technique.

#### General Procedure for Preparation of Schiff Bases (**4a**–**d**)

1 mmol of 4-(4-fluoroanilino)-6-methyl-2-phenylpyrimidine-5-carbaldehyde (**3**) [43] was dissolved in 10 mL of THF, then 1.5 mmol of aromatic amine and 2 mg of indium(III) trifluoromethanesulphonate were added. The mixture was stirred for 72 h at ambient temperature. After this time, the solvent was removed under a vacuum, and the remaining solid was dissolved in 25 mL of CHCl_3_ and then was washed with 25 mL of 2% aqueous HCl, dried over 5 g of MgSO_4_, and concentrated under a vacuum. The crude product was purified by column chromatography using CHCl_3_ as eluent.

*N-(4-fluorophenyl)-5-{[(4-fluorophenyl)imino]methyl}-6-methyl-2-phenylpyrimidin-4-amine* (**4a**). Product characterization: yield 0.30 g, 75.00%; yellow solid; melting point 178 °C; ^1^H NMR (300 MHz, CDCl_3_): δ (ppm) 2.83 (3H, s, CH_3_), 7.10–8.50 (13H, m, aromatic), 8.92 (1H, s, CH), 12.55 (1H, broad, NH). ^13^C NMR (151 MHz, CDCl_3_): δ (ppm) 162.58, 160.94′ (CAr-F), 160.19, 158.57′ (C_Ar_-F), 158.27 (CH=N), 156.69 (2C), 146.64, 135.08, 131.36 (2C), 128.96, 128.65 (4C), 123.59, 123.54, 122.70, 122.65, 116.54, 116.39, 115.76, 115.61, 107.23 (CAr), 22.86 (CH3). ^19^F NMR 471 MHz) δ (ppm) −62.71 (C-CF3), −115.80 (C_Ar_-F). HR-ESI-MS [M + H]^+^: found *m*/*z*: 401.1560, calcd. *m*/*z*: 401.1572 [mass error: 2.99 ppm]. FT–IR (ATR, selected lines): ν (cm^−1^) 1614 (C=N).

*5-{[(3-chlorophenyl)imino]methyl}-N-(4-fluorophenyl)-6-methyl-2-phenylpyrimidin-4-amine* (**4b**). Product characterization: yield 0.12 g, 28.85%; yellow solid; melting point 193 °C; ^1^H NMR (300 MHz, CDCl_3_): δ (ppm) 2.81 (3H, s, CH_3_), 7.10–8.49 (13H, m, aromatic), 8.91 (1H, s, CH), 12.38 (1H, broad, NH). ^13^C NMR (151 MHz, CDCl_3_): δ (ppm) 160.25, 158.64′ (C_Ar_-F), 158.32 (CH=N), 157.93 (2C), 151.83, 135.28, 131.46, 130.66 (2C), 129.00 (2C), 128.66 (2C), 126.76, 123.71, 123.66, 121.40, 119.70 (2C), 115.77, 115.62, 107.10 (C_Ar_), 22.07 (CH_3_). ^19^F NMR (471 MHz): δ (ppm) −117.54 (C_Ar_-F). HR-ESI-MS [M + H]^+^: found *m*/*z*: 417.1261, calcd. *m*/*z*: 417.1277 [mass error: 3.84 ppm]. FT–IR (ATR, selected lines): ν (cm^−1^) 1610 (C=N).

*N-(4-fluorophenyl)-6-methyl-2-phenyl-5-{[(2-methoxyphenyl)imino]methyl}pyrimidin-4-amine* (**4c**). Product characterization: yield 0.15 g, 36.41%; yellow solid; melting point 167–169 °C; ^1^H NMR (300 MHz, CDCl_3_): δ (ppm) 2.81 (3H, s, CH_3_), 3.94 (3H, s, CH_3_), 7.05–8.50 (13H, m, aromatic), 9.05 (1H, s, CH), 13.21 (1H, broad, NH). ^13^C NMR (151 MHz, CDCl_3_): δ (ppm) 172.43 (CH=N), 166.45 (C_Ar_), 160.59, 158.97′ (C_Ar_-F), 159.00 (2C), 153.50, 137.18, 134.02, 131.99 (2C), 129.38 (2C), 128.69 (2C), 124.30, 124.24 (2C), 121.37, 118.85, 115.80, 115.65, 112.01, 108.56 (C_Ar_), 55.96 (O-CH_3_), 21.57 (CH_3_). ^19^F NMR (471 MHz): δ (ppm) −117.62 (C_Ar_-F). HR-ESI-MS [M + H]^+^: found *m*/*z*: 413.1760, calcd. *m*/*z* 413.1772 [mass error: 2.90 ppm]. FT–IR (ATR, selected lines): ν (cm^−1^) 1634 (C=N).

*5-[{[4-chloro-3-(trifluoromethyl)phenyl]imino}methyl]-N-(4-fluorophenyl)-6-methyl-2-phenylpyrimidin-4-amine* (**4d**). Product characterization: yield 0.24 g, 51.65%; yellow solid; melting point 210 °C; ^1^H NMR (300 MHz, CDCl_3_): δ (ppm) 2.81 (3H, s, CH_3_), 7.10–8.48 (12H, m, aromatic), 8.91 (1H, s, CH), 12.17 (1H, broad, NH). ^13^C NMR (151 MHz, CDCl_3_): δ (ppm) 172.42 (CH=N), 166.44 (C_Ar_), 160.59, 158.99′ (C_Ar_-F), 158.69, 158.35, 149.38, 134.04, 132.73, 131.58, 129.92–129.30 (q, C_Ar_-CF_3_), 129.38, 129.06 (2C), 128.68 (2C), 125.22, 124.29, 124.23, 123.91, 123.86′ (C_Ar_-Cl), 123.60, 121.79′ (CF_3_), 120.71, 120.67′, 115.82, 115.68, 106.98, 21.56 (CH_3_). ^19^F NMR (471 MHz): δ (ppm) −62.67 (CF_3_), −117.53 (C_Ar_-F). HR-ESI-MS [M + H]^+^: found *m*/*z*: 485.1150, calcd. *m*/*z* 485.1151 [mass error: 0.21 ppm]. FT–IR (ATR, selected lines): ν (cm^−1^) 1646 (C=N).

### 3.2. X-ray Structural Studies

The diffraction data were collected on a XtaLAB Synergy R (DW system, Hy-Pix-Arc 150) [CuK_α_ = 1.5418 Å] or Xcalibur, Ruby (Gemini ultra) [MoK_α_ = 0.71073 Å] diffractometer at 100 K. The CrysAlisPro software package [68] was used for data collection, cell refinement, data reduction, and analysis. All crystal structures were solved by direct methods using SHELXS-97 [69] and refined by a full-matrix least-squares technique on F2 with SHELXL-2013 (and further with SHELXL-2018) [70]. All non-H atoms were refined with anisotropic displacement parameters. Crystal data, data collection, and structure refinement details are summarized in Table 1. All H atoms (except those –NH– groups, which could potentially be engaged in hydrogen bonds) were geometrically optimized and allowed for as riding atoms, with C–H distances in the range of 0.95–0.99 Å with U_iso_(H) = 1.2U_eq_(C). In all the studied structures, the methyl H atoms were refined with C–H = 0.98 Å and U_iso_(H) = 1.5U_eq_(C). Structure **4c** was refined as disordered in two positions with an occupancy factor of 0.5. In both molecules, the disordered part includes atoms from the (4-fluorophenyl)amino group attached to the C4 atom of the pyrimidine ring (disordered atoms are denoted as 42C-43C, 45C-46C in A and as 42D-46D and F4D in B, respectively). Moreover, in molecule B, the carbon atom of the methoxy group is also disordered in two positions, denoted as C58B and C58D. The structures were drawn using the DIAMOND [71] and MERCURY CSD 3.1 programmes [72].

CCDC 2322460-2322463 contains the supplementary crystallographic data for this article. These data can be obtained free of charge from the Cambridge Crystallographic Data Centre at www.ccdc.cam.ac.uk/data_request/cif (accessed on 7 September 2022).

### 3.3. Biological Activity Assays

#### 3.3.1. Materials

Dulbecco’s modified Eagle’s medium (DMEM), Eagle’s minimum essential medium (EMEM), Ham’s nutrient mixture F12, Williams’ medium E, foetal bovine serum (FBS), Dulbecco’s phosphate-buffered saline (PBS), L-glutamine solution, trypsin-EDTA solution, penicillin–streptomycin, amphotericin B solution, MEM non-essential amino acid solution, neutral red solution (3.3 g/L), propidium iodide (PI) solution, fluorescein diacetate (FDA), staurosporine, Hoechst 33342, levofloxacin, gentamicin, and amphotericin B were provided by Merck (Darmstadt, Germany). Dimethyl sulphoxide (DMSO) was obtained from ALCHEM (Toruń, Poland). Alexa Fluor 488-annexin V with annexin-binding buffer was purchased from Thermo Fisher Scientific (Waltham, MA, USA). Tryptone soya broth (TSB) and tryptone soya agar (TSA) were provided by Oxoid (Basingstoke, UK). Triphenyl tetrazolium chloride (TTC) was obtained from Alfa Aesar (Ward Hill, MA, USA).

L-929 (mouse C3H/An connective tissue), Caco-2 (human Caucasian colon adenocarcinoma), A172 (human glioblastoma), AGS (human Caucasian gastric adenocarcinoma), HepaRG (human hepatoma cell), and HeLa (human cervix epithelioid carcinoma) cell lines were obtained from the European Collection of Authenticated Cell Cultures (ECACC) and provided by Sigma Aldrich/Merck (Munich, Germany). All cell lines were cultured in complete mediums prepared according to the ECACC recommendations.

*Escherichia coli* 25922, *Staphylococcus aureus* 43300, *Klebsiella pneumoniae* 700603, *Acinetobacter baumannii* 19606, *Pseudomonas aeruginosa* 27853, *Enterococcus faecalis* 29212, and *Candida albicans* 10231 were purchased from the American Type Culture Collection (ATCC).

#### 3.3.2. Neutral Red Uptake Assay

Cell lines were plated in 96-well plates at a density of 1 × 10^4^ cells/well. After 24 h incubation, which allowed the cells to attach and start to proliferate, the culture media were replaced with **4a**–**d** dissolved in the appropriate medium at concentrations of 1, 10, 50, 100, 150 and 200 µM. DMSO did not exceed 1% in any of the test set. Negative (1% DMSO in medium) and positive (1 µM staurosporine) controls were also included. Plates were incubated at 37 °C for 24, 48, and 72 h before the neutral red uptake assay. The neutral red uptake assay was performed as described by Repetto et al. [73]. Briefly, the medium from all wells was discarded, the attached cells were washed with 100 µL PBS, and then 100 µL of neutral red working solution in medium (40 µg/mL) was added to each well. The neutral red solution was incubated overnight in the same conditions as the cells and centrifuged before being added to the cells. The plates were incubated for at least 2 h at 37 °C, 5% CO_2_, in a humidified atmosphere. Next, the neutral red medium was removed, and the cells were washed with 100 µL PBS. A 100 µL quantity of destain solution consisting of 50% ethanol, 1% glacial acetic acid, and deionized water was added to each well. Plates were incubated for 30 min with constant shaking on a horizontal shaker at 37 °C. Absorption was measured at 540 nm excitation. The cells’ viability was calculated by comparing results from experimental sets with negative controls according to the formula:$$\%\;viability=\frac{{OD}_{test\;substance}-{OD}_{blind}}{{OD}_{negative\;control}-{OD}_{blind}}\times 100$$

#### 3.3.3. Flow Cytometry

Two approaches for flow cytometry were used—FDA/PI staining to determine cell viability and annexin V/PI staining for cell death pathway determination.

For cell viability determination, cells were plated in 6-well plates at a density of 1 × 10^5^ cells/well and incubated for 24 h at 37 °C, 5% CO_2_ and in a humidified atmosphere. Next, cells were treated with **4b** at a concentration calculated as IC_50_ in neutral red uptake assay. After 24, 48, and 72 h incubation, the cells (either adhered or floating) were collected, centrifuged at 1600 rpm, and washed with PBS. FDA and PI stock solution at a 1 mM concentration were prepared in DMSO and H_2_O, respectively. After centrifugation and supernatant discarding, cells were stained with FDA at a final concentration of 1 µM prepared in PBS and incubated in the dark. After 15 min, they were centrifuged at 1600 rpm, resuspended in ice-cold PBS, and placed on ice. Samples were stained with 1 µM PI for 2 min before the cytometry evaluation. The sample acquisition was performed on a cyFlow Space flow cytometer, and matching FloMax software (ver. 2.9) was used for data analysis.

For cell death pathway determination, cells were plated in 6-well plates at a density of 3 × 10^5^ cells/well. Followed by 24 h incubation at 37 °C, 5% CO_2_, in a humidified atmosphere, cells were treated with **4b** at a concentration calculated as IC_50_ in neutral red uptake assay. After 6, 18, and 24 h incubation, adhered and floating cells were harvested, centrifuged at 1600 rpm, and washed with PBS. Next, the cells were washed with 1× annexin-binding buffer, and after centrifugation, cells from each test set were suspended in 100 µL of 1× annexin-binding buffer. To each tube, 5 µL of Alexa Fluor 488-annexin V stain was added, and the cells were incubated in the dark. After 15 min incubation, the cells were centrifuged at 1600 rpm to remove the dye, then suspended in 1 mL of 1× binding buffer and kept on ice. Prior to analysis, cells were stained for 2 min with 1 µM PI. The sample acquisition was performed on a cyFlow Space flow cytometer, and matching FloMax software was used for data analysis.

#### 3.3.4. Microscopic Observations

To visualize morphological changes which occurred in the treated cells, microscopic observations were conducted using Hoechst 33342 stain. It is a blue fluorescent stain which allows observations of nuclei in eukaryotic cells. Also observations in bright fields were performed. Cells were stained according to the manufacturer’s protocol. Briefly, cells were plated in 12-well plates at a density of 1 × 10^5^ cells/well and were incubated for 24 h at 37 °C, 5% CO_2_ in a humidified atmosphere. Next, cells were treated with **4b** at an IC_50_ and further incubated for 24, 48, and 72 h. For microscopic observations cells were collected, washed with PBS and stained for 15 min with 1 µg/mL Hoechst 33342 dye. Observations were conducted under inverted microscope with ultraviolet excitation filter (IX53, Olympus, Tokyo, Japan).

#### 3.3.5. Comet Assay

The comet assay was performed according to the method of Singh et al. [74] with slight modifications. Slides sterilized in 70% ethanol microscope were covered with thin layer of 1% normal melting agarose and left to solidify in a low-humidity environment. Cells were plated in 12-well plates at a density of 1 × 10^5^ cells/well and incubated for 24 h in a cell incubator. Next, compound **4b** was added in concentrations equal to the IC_50_ values obtained after the neutral red uptake assay. After 24, 48, and 72 h incubation, cells were harvested, centrifuged, and resuspended in PBS. The cell suspension was mixed with 1% LMPA (low melting point agarose), and then loaded onto microscope slides pre-coated with normal melting agarose. Cover slides were placed on the base slides, and the complete slides were left to solidify for 10 min at 4 °C. After this solidification time, the cover slides were removed, and the slides with cells were immersed for 1 h in chilled lysing solution containing 2.5 M NaCl, 10 mM Tris-HCl, 100 mM EDTA, and added just before use, 1% Triton X-100 pH 10. The slides were then transferred to a horizontal electrophoresis tank filled with an alkaline electrophoresis buffer (300 mM NaOH, 1 mM EDTA, pH 13). DNA electrophoresis was performed at 25 V, 300 mA for 20 min. Slides were then immersed in neutralizing buffer containing 0.4 M Tris-HCl, pH 10. After 10 min, the slides were washed gently with deionized water and left to dry. The cells’ DNA was stained with Hoechst 33342 dye. Observations were made using an inverted microscope with an ultraviolet excitation filter (IX53, Olympus). A total of 100 randomly selected cells were scored, and captured comets were classified based on the comet’s tail length as follows: class 0 contained comets with no damaged DNA; class 1, those with low damage; class 2, those with medium damage; class 3, those with high damage; and class 4, those with very high damage. The damage index (*DI*) was calculated as follows:$$DI=\left[\left(0\times n_{0}\right)+\left(1\times n_{1}\right)+\left(2\times n_{2}\right)+\left(3\times n_{3}\right)+\left(4\times n_{4}\right)\right],$$
$n_{0,1,2,3,4}$—the amount of comets in each class.

#### 3.3.6. Antimicrobial Activity Assay

Seven reference strains from the ATCC collection (*Escherichia coli* 25922, *Staphylococcus aureus* 43300, *Klebsiella pneumoniae* 13883, *Acinetobacter baumannii* 19606, *Pseudomonas aeruginosa* 27853, *Enterococcus faecalis* 29212, and *Candida albicans* 10231) were used. The assay was performed according to a modified version of Richard et al.’s protocol [75,76,77] and ISO standards 20776-1:2019 and 16256:2021 [78,79]. Microdilution with spectrophotometric measurement was used to calculate MIC_50_ (minimal inhibitory concentration which decreased the microorganisms’ viability to 50%) values. Briefly, compounds were prepared 100× concentrated in DMSO, and during the whole experiment, they were diluted to 1× with TSB. Serial dilutions of tested compounds were made directly on plates at a concentration range of 0.5 to 256 µg/mL. Microorganisms were incubated for 24 h at 37 °C (bacteria) or 25 °C (yeast) on Petri dishes with TSA. The bacteria/yeast suspensions were prepared at a density 5 × 10^5^ CFU/mL (for bacteria) or 0.5–2.5 × 10^5^ CFU/mL (for yeast) with TSB. Also, appropriate controls were contained in the experiment: the positive control consisted of a specific strain in TSB, and the negative control was TSB alone. To ensure that used strains were not modified and were still responding to antibacterial/antifungal compounds established by EUCAST, additional tests were conducted with levofloxacin, gentamicin, and amphotericin B. Microplates were incubated on a shaker for 24 h at 37 ± 1 °C or 25 ± 1 °C for the bacteria and yeast strains, respectively. After the incubation, spectrophotometric measurement was performed at 580 nm. MIC_50_ was calculated by comparing the results obtained for experimental sets with positive controls. Next, 50 µL aliquots of 1% (*m*/*v*) TTC solution were added to each well. The plates were incubated on the shaker for 24 h at 37/25 °C. In living microbial cells, colourless TTC is converted into red formazan crystals, which allows for visual observation and MBC/MFC (minimal bactericidal/fungicidal concentration) value assessment.

### 3.4. In Silico Analysis

#### 3.4.1. ADME Prediction Analysis

ADME prediction analysis of compounds **4b** and 5-[(4-ethoxyphenyl)imino]methyl-N-(4-fluorophenyl)-6-methyl-2-phenylpyrimidine-4-amine [43] was performed via SwissADME [80], a freely available software programme provided by the Swiss Institute of Bioinformatics. The software provides information about estimated predictors, such as basic physiochemical properties, lipophilicity, water solubility, pharmacokinetics, drug-likeness, and medicinal chemistry. Physiochemical properties are relevant due to their crossing of biological barriers. Lipophilicity is very important for pharmacokinetic drug discovery. Orally active drugs should consist of no more than 5 hydrogen bond donors and fewer than 10 hydrogen bond acceptors and have the molecular weight of less than 500 Daltons and an *n*-octanol and water partition coefficient of less than 5 [81]. Water solubility is connected to the oral admission of the drug. Water solubility is the major property of medicine absorption. Identification of the permeability of glycoprotein substrate is a predictor of pharmacokinetics. Knowledge about the interaction of the compound and cytochromes P450 gives information about effective drug elimination through metabolic biotransformation. The last predictor, as in medicinal chemistry, supports structural drug discovery. It informs us about the presence of problematic structural fragments inside the investigated compound.

#### 3.4.2. Molecular Docking Analysis

Molecular docking was performed using AutoDock Vina [82]. Input sequences were prepared by AutoDock Tools software 1.5.6 [83]. The structures of Schiff base (**III**) [43] and compound **4b** were prepared using Avogadro, version 1.2.0, which is an open-source molecular builder and visualization tool [84].

The structures used in docking analysis were optimized by the energy minimization in the MMFF94 force field [85]. Water molecules were removed from receptors, and polar hydrogens were added; as well, missing atoms were repaired. ADT 1.5.6 was used to investigate the activity in terms of binding affinity (Kcal/mol). The docking outcomes, e.g., bonds between ligand and receptor, and the binding affinity score for best-docked conformation are compared for reference antibiotics and analysed peptides and presented in Section 2.4.2.

## 4. Conclusions

The structures of obtained compounds were mainly determined by spectroscopic and single-crystal X-ray analysis. Similar to the Schiff base (**III**), which contains the 4-ethoxyphenyl substituent in the 5-position [43], the conformation of the molecule of the studied compounds is stabilized by the intramolecular N–H∙∙∙N hydrogen bond involving the H atom of an amine group and the N imine atom. It should be noted that the aryl substituent in the 5-position is the most twisted relative to the pyrimidine ring plane in **4b** (almost by 42°), which contains the 3-chlorophenyl group. In other compounds, the conformation of molecules is almost syn-planar, with the greatest interplanar distortion angles involving the aryl substituent at the 5-position in the range 5–15°. Application of the single-crystal X-ray method allowed us to confirm the presence of C–H∙∙∙F hydrogen bonds, as well as observe the other weak interactions, which have also been recognized before in similar 6-methyl-2-phenylpyrimidine derivatives [41,42,43]. In all crystalline forms, the fluorine atoms at the 4-position (as an acceptor) and the proton (H) of the methyl or aryl ring are engaged in the H-bonds (H∙∙∙F).

The cytotoxic effect was analysed using a regular cell line (RPTEC—renal proximal tubule epithelial kidney cells) and selected reference cancer cell lines (A172—glioblastoma, AGS—gastric adenocarcinoma, HepG2—hepatocyte carcinoma, Caco-2—colon adenocarcinoma and HeLa—epithelioid cervix carcinoma) from ECACC (the European Collection of Cell Cultures). Results showed that the highest cytotoxicity against RPTEC has been found in **4a** (65% in 250 μM after 72 h) and **4c** (67% in 250 μM after 72 h).

However, the highest cytotoxic effect (<100 µM) has been determined for **4b**, which contains the 3-chlorophenyl substituent in the 5-position. For AGS and A172, IC_50_ = 32.2 μM and IC_50_ = 63.4 μM were calculated, respectively. Moreover, compound **4d** was toxic for AGS (IC_50_ = 189.5 μM). No relevant cytotoxicity against cancer cell lines was determined for the other compounds. The results obtained for compound **4b** have been compared with previously published results regarding the selected imine (compound **III**, Figure 1) [43]. The cytotoxic effect was increased in **4b** in comparison with **III** (IC_50_ equal to 32.2 µM and 50.2 µM for AGS and IC_50_ equal to 63.4 µM and 526.2 µM for A172, respectively) [43]. Nevertheless, neither MIC nor MBC/MFC activity has been determined for the concentration from 0.5 µg/mL to 256 µg/mL for any of the tested compounds, whereas for compound **III** [43] the antimicrobial activity was determined against *E. faecalis* (MIC = 16 µg/mL and MBC = 32 µg/mL) next to anticancer activity.

In order to assess the potential of the obtained compound, the results were compared against the gold standards for the treatment of glioblastoma multiforme (GBM), a gastric adenocarcinoma. For GBM, the initial chemotherapy involves the use of temozolomide (TMZ), a derivative of dacarbazine. The authors of the publication [86] incubated the A172 cell line for 72 h with 200 µM TMZ, resulting in a reduction of cell viability to just below 70%. Apart from the poor efficacy of TMZ, usage of it is also associated with numerous side effects.

In the case of gastric adenocarcinoma, adjuvant therapy following resection involves a combination of 5-fluorouracil, cisplatin, and epirubicin. Suttie et al. [87] determined the LD50 values for the AGS cell line, which were 153.7 µM for 5-fluorouracil (5-FU), 16.7 µM for cisplatin, and 0.23 µM for epirubicin. High concentrations of 5-FU significantly impact the occurrence of side effects, greatly affecting the well-being of patients [88].

Considering the several-fold lower IC_50_ values obtained for the derivative described in the present publication, our results may serve as a basis for alternative and more effective cancer treatment strategies. According to ADME prediction analysis, compound 4b has a chance to be a poorly soluble oral drug. VEGFR2 receptor is a promising target for new therapies reaching apoptosis-dependent cancers. An estimation of the binding site obtained during molecular docking to VEGFR2 receptor revealed that compound 4b has a better binding affinity to the receptor than the previously analysed Schiff base III [43]. Moreover, the other docking outcomes, e.g., bonds formed between the receptor and ligand, show the possible connections to VEGFR2 receptor.

To summarize, the obtained results for **4b** are very encouraging and should be useful in anticancer drug discovery.

## Data Availability

Data are contained within the article or Appendix A.

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
