# Peer review of "Pyrimidine Schiff Bases: Synthesis, Structural Characterization and Recent Studies on Biological Activities"

_ijms, 2024, doi:10.3390/ijms25042076_

Round 1

Reviewer 1 Report

Comments and Suggestions for Authors

I cannot recommend the publication of this work in International Journal of Molecular Sciences. The chemical content of the article is very weak. The work requires serious improvement.

1. The authors describe only 4 new compounds with a random set of substituents. The motivation for just such a small set of substituents in the series of aromatic amines must be justified.

2. The authors should optimize the reaction for the formation of Schiff bases, since for compounds 4b and 4c the yields are terribly small, only 29 and 36, respectively. The authors should also pay attention to the fact that yields are usually given to whole numbers, since it is hardly possible to achieve reproducibility down to tenths, much less up to hundredths.

3. It is common for high-ranking journals to include 13C NMR spectra in their Supporting Materials file. Moreover, since all the obtained compounds contain fluorinated substituents, it is necessary to provide 19F spectra.

4. The 1H NMR spectra for compounds 4a-e (see Supporting Materials file) show the obvious presence of impurities. The authors need to provide data on the purity of the obtained compounds according to HPLC data.

5. It is not clear on what basis the authors consider compound 4b to be a promising anticancer compound if the standard (any known anticancer drug) is not given in Table 4.

6. The article mentions the assessment of antibacterial activity for compound 4b, while it is not considered for other products. In addition, there is no section in the text that discusses the results of antibacterial activities.

Author Response

Dear Reviewer:

Thank you for the reviewing our contribution and giving us possibility to improve our manuscript in the best possible way.

I would like to thank you for reviewing the manuscript and providing constructive suggestions on our manuscript.

I have studied all the comments carefully and tried my best to revise the manuscript according to your valuable comments.

Any revisions to the manuscript were marked up using the “Track Changes” function if you are using MS Word/LaTeX, such that any changes can be easily viewed by the editors and reviewers. Additionally, all changes were highlighted in yellow (for Reviewer 1), turquoise (for Reviewer 2) or green (corrections of English language or other errors).

Below are the detailed responses (in red) to all points:

Response to Reviewer #1

Comments: “I cannot recommend the publication of this work in International Journal of Molecular Sciences. The chemical content of the article is very weak. The work requires serious improvement.”

-Authors’ reply: We are very sorry that our current experimental design did not fully convince you.

  1. The authors describe only 4 new compounds with a random set of substituents. The motivation for just such a small set of substituents in the series of aromatic amines must be justified.

-

We would like to thank the reviewer for this valuable comment. Moreover, we understand the reviewer's point of view. However, this randomness of substituents was well thought out. Our goal was to obtain new compounds showing biological activities and thus find out the basis for the design of new compounds with better biological properties. We intended to obtain only predicted structure.

Additionally, because we suspected that the introduction of a 4-fluorophenyl moiety at the 4-position of the pyrimidine had an impact on the increase of  anti-cancer activity compared to previously tested 2-phenyl-6-methylpyrimidine systems, we made the smallest possible structural changes, retaining the skeleton of compound III (see Schemat 1). We  limited counts of compounds. In this way the structural elements of this molecule were exchanged in the following directions and only in the 5-position of the pyrimidine ring:  a change of the substituent and/or the introduction of an additional substituent with different electro-/nucleophilic properties (both) to the aryl ring connected to the imine group (and also in different positon of this aryl ring) (Scheme 1 ).

In this way, to coupling with aldehyde (3) (Scheme 2) we selected amines with different basicity. In particularly, substituted arylamines can be either more basic or less basic than aniline, depending on the substituent. Electron-donating substituents, such as  -OCH3 (4c) (in comparison to -OC2H5 of compund III), or -CF3 (4d) increase the basicity of the corresponding arylamine. Electron-withdrawing substituents, such as -F (4a) or -Cl (4b i 4d), decrease arylamine basicity.

Moreover, we focused on a broader X-ray studies and biological aspect, which also involves more time and a large financial outlay, hence the limitation on the number of compounds tested in this way.

  1. The authors should optimize the reaction for the formation of Schiff bases, since for compounds 4b and 4c the yields are terribly small, only 29 and 36, respectively.

- We agree with this comment. Final Schiff bases 4a-d were obtained by coupling aldehydes (3) with aromatic amines in THF in the presence of a catalytic amount of indium(III) trifluoromethanesulfonate (In(OTf)3) catalyst, for 72 hours at ambient temperature. The reaction conditions were the same for these compounds, which indicated the weak points of this approach.

However, we have not optimized the reaction conditions for the formation of these Schiff bases. We intend to do this in the next step.

The authors should also pay attention to the fact that yields are usually given to whole numbers, since it is hardly possible to achieve reproducibility down to tenths, much less up to hundredths.

- This part has been corrected.

  1. It is common for high-ranking journals to include 13C NMR spectra in their Supporting Materials file. Moreover, since all the obtained compounds contain fluorinated substituents, it is necessary to provide 19F spectra.

-In response to your request, we have prepared a Supplementary file containing the 13C and 19F NMR spectra used to elucidate the structure of studied compounds. While all the identified molecules were also clearly determined in a single-crystal X-ray analysis, we believe that providing NMR spectra as supplementary material will enhance the transparency and reliability of our work.

  1. The 1H NMR spectra for compounds 4a-e (see Supporting Materials file) show the obvious presence of impurities. The authors need to provide data on the purity of the obtained compounds according to HPLC data.

- We would like to thank the reviewer for this valuable comment.

Previous 1H NMR spectra were included inadvertently - the spectra presented in the supplemental materials (prior to review) were performed before final purification of the compounds. We are very sorry for the disruption.

Moreover, we understand the reviewer's point of view, but it is very difficult to conduct such experiments in the few days we have to review the article.

Most of the samples were used for biological tests. We had only a very small amount of each of sample, which used to HR-ESI-MS spectra to confirm the purity of tested compounds.

  1. It is not clear on what basis the authors consider compound 4b to be a promising anticancer compound if the standard (any known anticancer drug) is not given in Table 4.

- We thank the reviewer for this valuable comment. Table 4 includes the human cell lines treatment with tested compounds with. In section 2.4 The authors performed in silico studies analyzing ADME parameters to assess the pharmacokinetic profile of Schiff bases. Techniques such as in vitro and in vivo studies, as well as computational methods, are employed to gather data on these ADME parameters. It's worth noting that the specific ADME characteristics of Schiff bases can vary in term of  their chemical structure, substituents, and the presence of functional groups that can influence these parameters. Specific examples of pyrimidine Schiff bases that were under clinical trials for anticancer purposes were not readily available until today. ADME analysis focuses on comparison of studied compound 4b and previously published compound III. We have also added another example of a biological active compound found using a SwissSimilarity database [46] with the best similarity score to compound 4b.

[46] Zoete, V., Daina, A., Bovigny, C., & Michielin, O. SwissSimilarity: A Web Tool for Low to Ultra High Throughput Ligand-Based Virtual Screening., J. Chem. Inf. Model., 2016, 56(8), 1399-1404.

  1. The article mentions the assessment of antibacterial activity for compound 4b, while it is not considered for other products. In addition, there is no section in the text that discusses the results of antibacterial activities.

- The assessment of antibacterial activity is present in part 2.3.5. Additionally, the entire passage of text has been rephrased in Conclusion. The line regarding antimicrobial assay has been modified to emphasize that within the specified concentration range, all compounds were tested, not just 4b. 

Reviewer 2 Report

Comments and Suggestions for Authors

Although this study does not deal with synthesis of many compounds (only 4 are synthesized with intermediates being already reported by the authors) it is experimentally rich. The work is highly multidisciplinary-  there is important information on X_ray structures and a well-developed biological context (in vitro and in silico). Hence, I recommend publication of this manuscript after some substantial issues are resolved/commented by the authors:

1)     In the introduction the authors summarize their own work starting form compound II. What would be compound I then? In addition, this discussion starts with 4-[(4-chlorobenzyl)sulfanyl]-5,6-dimethyl-2-phenylpyrimidine and its 5-hydroxymethyl analog. Why those are not included in scheme 1? Finally, I would depict in scheme 1 what Ar is in compounds 4a-d.

2)     End of the Introduction part: The sentence “(…)the X-ray single-crystal and ADME analyses were carried out’ is not entirely true. Please rewrite to “ADME prediction analyses”.

3)     Synthesis description - If “Details for the synthesis of 13 were described by the authors previously” and this synthesis is referenced , then it should not be described again here in the text and should be removed form Scheme 2

4)     Please remove the temperatures from the discussion of synthesis, as it is too technical. Instead, it would be more proper to include them in the caption to Scheme 2

5)     Scheme 2 – the authors may consider including chemical yields in the graph. Besides that, why the alternative nomenclature (ie. XIII,…) is used and why in red?

6)     Scheme 2 caption – for some transformations the reaction time is included but not for LAH reduction. Why?

7)     Supplementary information – please provide the full spectral range (0 to 15 ppm) for 1HNMR. I would also recommend to include 13C NMR spectra. The compounds are not of exceptionally high MWs so this should not be resource-consuming.

8)     The docking analysis is meaningless. There are three main reasons I think so. First, it appears sketchy. There is not even a 3D representation of binding. Then, more importantly, there could be numerous (thousands?) potential molecular targets for these compounds, so why to choose two only? In general, I fell the authors are convinced by the docking results, but those results are meaningless without experimental (in vitro) validation.

Comments on the Quality of English Language

1The use of English is fair and it allows to follow the manuscript and to understand its scientific content. However some sentences do have slightly awkward construction. I would recommend to further improve the language.

Author Response

Dear Reviewer:

Thank you for the reviewing our contribution and giving us possibility to improve our manuscript in the best possible way. I would like to thank you for reviewing the manuscript and providing constructive suggestions on our manuscript.

I have studied all the comments carefully and tried my best to revise the manuscript according to your valuable comments.

Any revisions to the manuscript were marked up using the “Track Changes” function if you are using MS Word/LaTeX, such that any changes can be easily viewed by the editors and reviewers. Additionally, all changes were highlighted in yellow (for Reviewer 1), turquoise (for Reviewer 2) or green (corrections of English language or other errors).

Below are the detailed responses (in red) to all points:

Response to Reviewer #2

Although this study does not deal with synthesis of many compounds (only 4 are synthesized with intermediates being already reported by the authors) it is experimentally rich. The work is highly multidisciplinary-  there is important information on X_ray structures and a well-developed biological context (in vitro and in silico). Hence, I recommend publication of this manuscript after some substantial issues are resolved/commented by the authors:

  • In the introduction the authors summarize their own work starting form compound II. What would be compound I then? In addition, this discussion starts with 4-[(4-chlorobenzyl)sulfanyl]-5,6-dimethyl-2-phenylpyrimidine and its 5-hydroxymethyl analog. Why those are not included in scheme 1? Finally, I would depict in scheme 1 what Ar is in compounds 4a-d.

We would like to thank the reviewer for these valuable comments. Mentioned compounds, i.e. 4-[(4-chlorobenzyl)sulfanyl]-5,6-dimethyl-2-phenylpyrimidine and its 5-hydroxymethyl analog, are shown in Scheme 1 as suggested. Scheme 1 also shows what Ar is in compounds 4a-d.

  • End of the Introduction part: The sentence “(…)the X-ray single-crystal and ADME analyses were carried out’ is not entirely true. Please rewrite to “ADME prediction analyses”.
  • We thank the reviewer for the comment. The sentence was rewritten according to suggestion.
  • Synthesis description - If “Details for the synthesis of 1–3 were described by the authors previously” and this synthesis is referenced , then it should not be described again here in the text and should be removed form Scheme 2
  • We agree with this comment. This part has been corrected.

4)     Please remove the temperatures from the discussion of synthesis, as it is too technical. Instead, it would be more proper to include them in the caption to Scheme 2.

  • We agree with this comment. This scheme has been corrected.
  • Scheme 2 – the authors may consider including chemical yields in the graph. Besides that, why the alternative nomenclature (ie. XIII,…) is used and why in red?
  • The alternative nomenclature (ie. XIII,…) was included inadvertently. We apologise for disturbance.
  • Scheme 2 caption – for some transformations the reaction time is included but not for LAH reduction. Why?
  • This part has been corrected.
  • Supplementary information – please provide the full spectral range (0 to 15 ppm) for 1HNMR. I would also recommend to include 13C NMR spectra. The compounds are not of exceptionally high MWs so this should not be resource-consuming.
  • This part has been corrected.

8)     The docking analysis is meaningless. There are three main reasons I think so. First, it appears sketchy. There is not even a 3D representation of binding. Then, more importantly, there could be numerous (thousands?) potential molecular targets for these compounds, so why to choose two only? In general, I fell the authors are convinced by the docking results, but those results are meaningless without experimental (in vitro) validation.

  • We thank reviewer for this valuable comment. The 3D visualization of compounds in receptor binding site was added to publication.

Comments on the Quality of English Language

1The use of English is fair and it allows to follow the manuscript and to understand its scientific content. However some sentences do have slightly awkward construction. I would recommend to further improve the language.

  • We agree with this comment. Most of the English language errors have been corrected.

Round 2

Reviewer 1 Report

Comments and Suggestions for Authors

The authors have forgotten to attach the corrected the Supporting Materials file.

Comments on the Quality of English Language

Minor editing of English language required.

Author Response

Dear Reviewer:

Thank you once again for reading our manuscript and giving us another opportunity to improve it in the best way possible.

Any revisions to the manuscript were marked up using the “Track Changes” function if you are using MS Word/LaTeX, such that any changes can be easily viewed by the editors and reviewers. Additionally, all changes introduced after review 1 remain highlighted in yellow (for Reviewer 1), turquoise (for Reviewer 2) or green (corrections of English language or other errors). New English corrections are highlighted in grey.

Response to Reviewer

C1:The authors have forgotten to attach the corrected the Supporting Materials file.

R1: I'm very sorry for the confusion. I have uploaded a revised "supplementary material" file, but it has not been included correctly on the website. I hope it will be visible now.

C2 (Comments on the Quality of English Language): Minor editing of English language required.

R2: New English corrections are highlighted in grey.

Round 3

Reviewer 1 Report

Comments and Suggestions for Authors

 Authors have carefully checked and modified this manuscript. Now it can be accepted for publication in this journal without further revision.